# Regime shifts occur disproportionately faster in larger ecosystems

Gregory S. Cooper [1,4], Simon Willcock [2,4] & John A. Dearing[3 ✉]

Regime shifts can abruptly affect hydrological, climatic and terrestrial systems, leading to degraded ecosystems and impoverished societies. While the frequency of regime shifts is predicted to increase, the fundamental relationships between the spatial-temporal scales of shifts and their underlying mechanisms are poorly understood. Here we analyse empirical data from terrestrial ($n = 4$), marine ($n = 25$) and freshwater ($n = 13$) environments and show positive sub-linear empirical relationships between the size and shift duration of systems. Each additional unit area of an ecosystem provides an increasingly smaller unit of time taken for that system to collapse, meaning that large systems tend to shift more slowly than small systems but disproportionately faster. We substantiate these findings with five computational models that reveal the importance of system structure in controlling shift duration. The findings imply that shifts in Earth ecosystems occur over 'human' timescales of years and decades, meaning the collapse of large vulnerable ecosystems, such as the Amazon rainforest and Caribbean coral reefs, may take only a few decades once triggered.

[1] Centre for Development, Environment and Policy (CeDEP), School of Oriental and African Studies, University of London, London WC1H 0XG, UK. [2] School of Natural Sciences, Bangor University, Bangor LL57 2DG, UK. [3] Geography and Environmental Science, University of Southampton, Southampton SO17 1BJ, UK. [4]These authors contributed equally: Gregory S. Cooper, Simon Willcock. ✉email: J.Dearing@soton.ac.uk

Anthropogenic activities are dependent upon the persistence of various biophysical conditions, such as soil fertility, freshwater availability and stable fish populations[1]. However, regime shifts can cause significant negative impacts on Earth's contemporary social–ecological systems[2]. For example, marine fishery collapses over the past 50 years have degraded continental-scale food securities and economic opportunities[3,4]. Such shifts also exist at local scales, with coastal lagoons, estuaries and freshwater environments susceptible to significant declines in ecosystem conditions and socioeconomic productivity[5]. Here, we conceptualise regime shifts as large, persistent, and often unexpected changes in relatively stable ecosystems[6,7], which may (or may not) be driven by reinforcing feedback loops beyond 'tipping points'[8,9]. From this definition, we consider shift duration to be the time taken to transition to a stable but functionally different system state[8].

Problematically for their governance, regimes shifts are traditionally viewed as abrupt relative to the temporal scales of the initial and resulting regimes[4,10]. Regime shifts are often associated with a preceding decline in resilience, associated with the inability of system structures to maintain stability under stress[11,12]. However, the current suite of resilience metrics currently lack robust cross-system transferability[13] and the general sparsity of quantitative information on regime shifts further complicates their prediction and governance[9]. With the frequency of regime shifts predicted to increase in association with climate change and environmental degradation[14], developing the general understanding into the spatial and temporal dynamics of shifts would help to anticipate the nature and timing of potential impacts; improve understanding into the role of system structure on resilience; and identify sizes of 'windows of opportunities'[15] to implement adaptive management to reduce socioeconomic and ecological damage.

We use network science to inform two hypotheses that link the speed of a regime shift to the size and structure of the system (Fig. 1). We hypothesize that larger systems (as measured by area) should intuitively take longer in absolute terms to shift between alternate regimes due to time-distance relationships, the diffusion of stresses and built-in time-lags. However, systems vary in terms of the speed by which a stressor may transmit through a system, from fluid, highly connected atmospheres and water bodies to less fluid terrestrial systems where physical infrastructure, like soil horizons and river channels, may reduce transmission speeds across the whole system. In turn, modularity, or the relative number of independent (i.e. unconnected) sub-systems, is a structural attribute that potentially slows cascading effects once a transition has been triggered (i.e. many independent smaller systems tip cumulatively more slowly than a single larger system of the same total size; Fig. 1)[16].

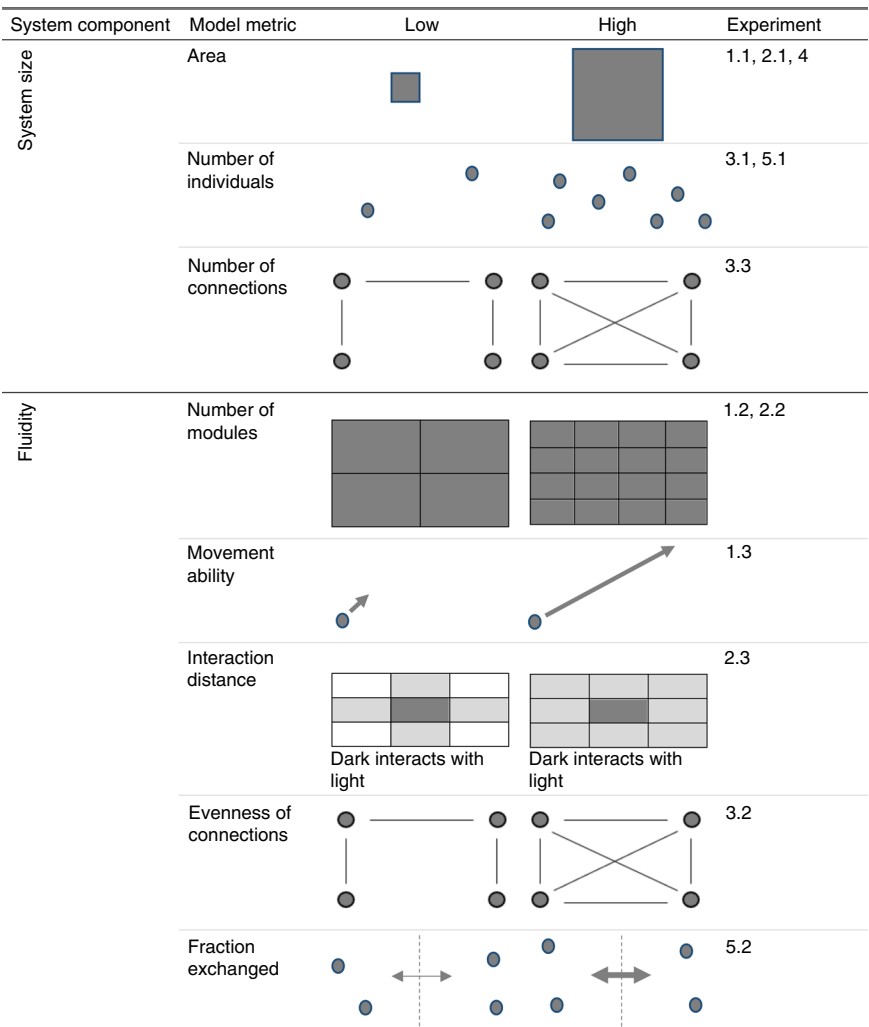

**Fig. 1 Graphical representation of the modelling framework.** Each row shows two graphics to illustrate the extreme variants (low, high) for a specific metric associated with either system size (upper) or fluidity (lower) in the 12 modelling experiments.

Hierarchical, self-organized biological systems and ecosystems possess attributes which scale sub-linearly with system size through the limits of energy dissipation, including tree branching or blood vessel networks[17], and production or predator-prey biomass[18]. In terms of network connectivity, it is known that heterogenous, hierarchical systems are resilient to random failure but vulnerable to targeted attack or failure of keystone nodes[19]. It follows that such systems should cascade relatively quickly once the keystone nodes are damaged or extirpated. In mechanistic terms, the breakdown of organisation during a regime shift might be expected to track the same sub-linear scaling trend. Thus, we also hypothesise that the size-duration relationship will tend to display a sub-linear power-law relationship — indicating relatively faster regime shifts for large systems.

To test these two hypotheses, we first compile empirical data on real-world ecosystem shifts from scientific publications, institutional reports and online collations such as the Regime Shifts Database[7] and the Threshold Database[20] (see Methods and Supplementary Table 1). Each of the 42 observed shifts meet various criteria for inclusion (see Methods) based upon the characteristics of the shift from one regime to another and our ability to precisely and reliably estimate the spatial and temporal extents of the shifts. Modelling has revealed the likely type of regime shift in some cases (i.e. the presence of a tipping point, critical transition or hysteresis[2,8]), but we make no assumptions about the reversibility of each shift. While we recognise that our empirical sample is not exhaustive, the dataset covers a variety of biophysical systems across seven orders of spatial magnitude, three orders of temporal magnitude, five continents and three environmental settings.

We substantiate these real-world relationships with five freely available computational models: Wolf-Sheep Predation (WSP)[21], Game of Life (GoL)[22], Language Change (LC)[23,24], Lake Chilika fishery (CHL)[25] and Spatial Heterogeneity (SH)[2] which show the potential spatial characteristics behind the empirical relationships (Table 1 and Methods). A total of twelve ecological modelling experiments were designed to unravel the hypothesised effects of scale, fluidity, modularity, and the heterogeneity of connections on the duration of regime shifts (Table 1 and Fig. 1). In particular, we selected models with dynamic variables that are capable of shifting from one state to another, and which we could control explicitly for either system size or system structure in multiple runs (see Methods). We selected models that capture both reversible, non-catastrophic shifts (e.g. WSP) and catastrophic shifts where reversibility demands overcoming hysteresis[26] (e.g. fishery collapse in the CHL model). Moreover, these models are all freely available making the experiments reproducible with the model details and NetLogo codes provided in Supplementary Tables 3–7.

We find positive sub-linear empirical relationships between the size and shift duration of systems for both the empirical and modelled data. This indicates that large systems tend to shift more slowly than small systems but faster per unit area. Using this relationship, we predict that the collapse of large vulnerable ecosystems (e.g. the Amazon rainforest) may take only a few decades once triggered.

## Results and discussion

**Empirical data**. As hypothesised, the real-world records show a positive association between system area and shift duration (Fig. 2), implying that shifts in larger systems, once triggered, take longer to reach a new regime. The overarching relationship is also sub-linear (Fig. 2), remaining statistically significant both with (slope = 0.221, $R^2 = 0.491$, $p < 0.001$, df = 40) and without the Sahara record (slope = 0.190, $R^2 = 0.423$, $p < 0.001$, df = 39).

We tested the robustness of this relationship with two sensitivity analysis experiments (as well as the plotting of generalised linear models, which can be found in Supplementary Table 1): (i) where alternative datasets were generated ($n = 42$), with one record from the original dataset removed in each (Supplementary Table 2), and (ii) using a Monte Carlo style approach where each of the original 42 records was given random error magnitudes between 50% and 150% of their original values across 5000 probabilistic simulations (see Methods and Supplementary Figs. 5–6). In the first sensitivity experiment, all of the 42 alternative models were found to have positive and sub-linear relationships between system size and shift duration (all significant to $p < 0.001$ level; Supplementary Table 2). Moreover, the original empirical model (Fig. 2) is found to be most sensitive to the record of the Sahara Desert; however, the removal of its record leads to a 14% decrease in the 'b-coefficient/slope term', making the new model more sub-linear. In the second sensitivity experiment, all 5000 simulations exhibited positive and sub-linear relationships between system size and shift duration (all significant to $p < 0.001$ level; Supplementary Fig. 6). Therefore, we infer that the power-law relationship is robust (Fig. 2), and not dependent on any one datapoint nor the assumption that the empirical dataset has unreasonably narrow error bounds. The robust sub-linear power-law relationships suggest that although there is an overarching positive association between area and shift duration, larger systems shift comparatively quickly relative to their size. In other words, the change in shift duration slows down as system size increases, implying that the trend line asymptotes towards some theoretical maximum shift time for Earth's ecosystems. A similar result is observed for estimated system volumes (Supplementary Fig. 3) and additional empirical data are unlikely to fill the size-duration space (Fig. 2) to the extent that the broad statistical relationships are overturned; instead, uncertainty surrounding this relationship would likely be reduced.

The empirical results provide first order estimations for the shift durations of iconic ecosystems like the Amazon rainforest and Caribbean coral reefs. The empirical model (Fig. 2) estimates that an ecosystem the size of the Amazon (~5.5-million km$^2$) will shift over 49 years (95% confidence interval [CI]: 10–260 years), which is broadly in line with the multidecadal shift durations projected by expert judgement[9] and process-based models[14,27]. Worryingly, recent plot inventories from the Amazon show a declining rate of carbon sequestration[28], and there is growing evidence that further deforestation and degradation of the feedback between moisture formation and vegetation coverage may lead to a system-wide tipping point as soon as 2021[29,30]. For a system the size of the Caribbean coral reefs (~20,000 km$^2$)[31], the empirical model estimates a 15 year period (95% CI: 5–50 years) to collapse once triggered. The decadal timescales are coherent with the observations that coral cover across the Caribbean declined by 80% from 1977 to 2001[32] and may completely disappear by 2035[33], depending on rates of further overfishing, climate change and ocean acidification. While the uncertainty bounds around the mean estimates must be acknowledged, these two collapses remain within 'human' timescales of years and decades (stretching to centuries at the edges of the 95% confidence interval).

**Model results**. First, we analyse the direct effects of system size on shift duration and whether the change in regime shift duration generally increases (super-linear) or decreases (sub-linear) with the change in spatial dimension (Fig. 3). Here, the modelled results are consistent with our empirical findings and show that shift duration is positively and sub-linearly associated with increasing system size as measured by system area (WSP-1.1, GoL-2.1 and CHL-4) and carrying capacity (SH-5.1) (Fig. 3; slope

**Table 1 Details and hypotheses of the 12 modelling experiments designed to substantiate the empirical relationship observed in Fig. 1.**

| Model name | Model type | Parameter varied (experiment num.) | Parameter range | Repeats per parameter value | Number of runs (n) | Hypothesis |
|---|---|---|---|---|---|---|
| Wolf-Sheep Predation (WSP) | Agent-based | (1.1) Model total area | World height: [0–100] World width: [0–100]. | 100 | 260,100 | Larger system areas should exhibit longer shift durations |
| | | (1.2) Module size (divide constant 100 × 100 area into sub-worlds) | [2, 5, 10, 20, 50, 100][a] | 100 | 60,600 | More modular systems should exhibit longer shift durations |
| | | (1.3) Maximum distance wolves and sheep can move per time step | [1–100 cells] | 100 | 10,000 | More fluid systems should exhibit shorter shift durations |
| Game of Life (GoL) | Cellular automata | (2.1) Model total area | World height: [0–100] World width: [0–100]. | 100 | 260,100 | Larger system areas should exhibit longer shift durations |
| | | (2.2) Module size (divide constant 100 ×100 area into discrete sub-worlds) | [2, 5, 10, 20, 50, 100] | 100 | 600 | More modular systems should exhibit longer shift durations |
| | | (2.3) Number of neighbouring cells any one cell can interact with | 4 or 8 | 100 | 200 | More fluid systems should exhibit shorter shift durations |
| Language Change (LC) | Network-structured | (3.1) Number of network nodes | [3–1000] | 100 | 99,800 | Networks with more nodes should exhibit longer shifts |
| | | (3.2) Number of inter-nodal connections | [99–4500] | 100 | 440,200 | Networks with more connections should exhibit longer shifts |
| | | (3.3) Standard deviation of connections measured from experiment 3.2. | [99–4500] | 100 | 440,200 | Networks with more homogenous connections should exhibit longer shifts |
| Lake Chilika (CHL) | System dynamics model | (4) Model total area | [500–10,000 km²] | 5000 areas randomly sampled between limits | 5000 | Larger system areas should exhibit longer shift durations |
| Spatial Heterogeneity (SH) | Ordinary differential equation | (5.1) Carrying capacity for phytoplankton (i.e. model size) | [1–100] | 1 (model does not have stochasticity) | 101 | Large systems should exhibit longer shift durations |
| | | (5.2) Fraction of volume exchanged between model parts (i.e. diffusion of stress) | [1–100] | 1 (model does not have stochasticity) | 101 | More fluid systems should exhibit shorter shift durations |

[a]The grass regrowth rate in experiment WSP-1.2 was also varied 1–100 and statistically controlled for in our regression models (see Methods). See Methods and Supplementary Notes 3–5 for additional and replicable details on the structure, parameterisation and code of the models.

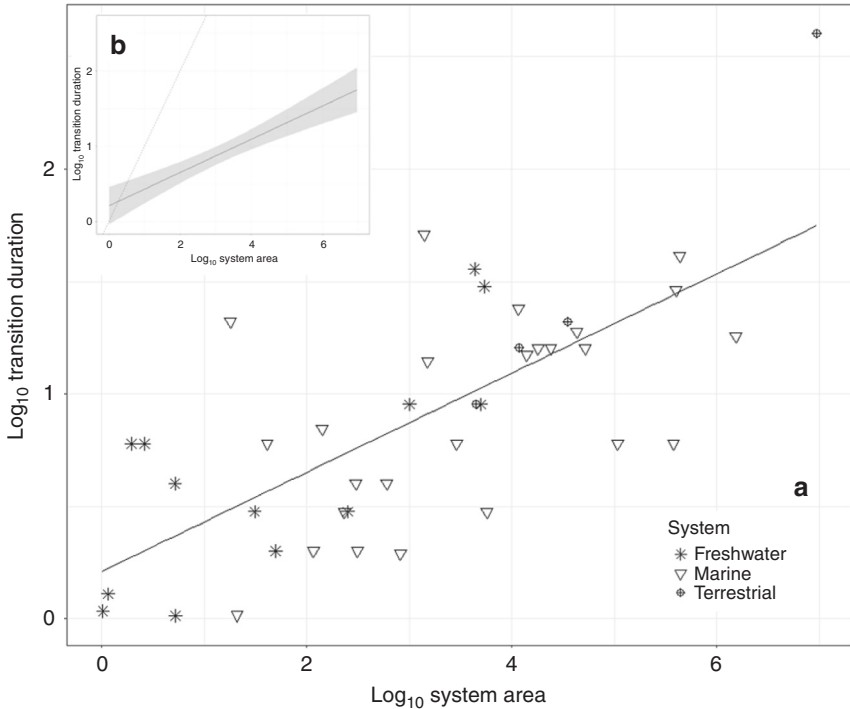

**Fig. 2 Empirical relationship between system area and regime shift duration. a** The log–log linear relationship between the spatial area and the temporal duration of 42 observed Earth system regime shifts is described by a linear regression model (solid line: $R^2 = 0.491$, $p < 0.001$, df $= 40$). This illustrates the positive and sub-linear (slope $= 0.221$) association between system size and shift duration. **b** The relationship in A is compared with the 1:1 reference line (dashed line, slope $= 1$). The untransformed unit of the x-axis is kilometres-squared, while the y-axis is years. The shading represents the 95% confidence interval around the regression model; see Supplementary Table 1 for individual case-study details and see Supplementary Fig. 4 for the regression models grouped by system type.

term, $b < 1$). The exception is where system size is measured by the number of system nodes (LC-3.1) and then shift duration scales super-linearly (slope $> 1$).

Second, we compare the model outputs with their different structures to gain insight into why the sub-linear relationships exist. Consistent with our hypothesis, we find that shift duration is negatively correlated with system fluidity in both the GoL (GoL-2.3) and the SH (SH-5.2) models (Fig. 3). We also find that shift duration is positively associated with system modularity (WSP-1.1 and GoL-2.2), where more modular (i.e. less fluid) systems are slower to shift from one regime to another. Moreover, heterogeneously wired systems (i.e. those with keystone nodes) tend to require less time to transition once a shift has been triggered (LC-3.3).

The two structural experiments that are inconsistent with our initial hypotheses are the WSP fluidity (WSP-1.3) and LC connections (LC-3.2) simulations as we find these had no effect on shift duration. Therefore, we hypothesise that the GoL fluidity experiment (GoL-2.3) suggests that spatial fluidity better influences regime shift duration when the direction of stress transmission is less restricted; for example, switching between a von Neumann four-direction neighbourhood[34] to a Moore eight-direction neighbourhood[34] (see Methods). In contrast, the insignificant WSP fluidity result suggests that the ability of a stressor to move further through a system is less important than the subsequent target of the stress because the stressor may just attack a resistant or unimportant part of a system in a distant location (e.g. a non-keystone node). This hypothesis is also supported empirically, though data are sparse (Supplementary Fig. 4), with the four terrestrial systems (less fluid) plotting along a steeper line than other systems containing freshwater and marine transitions (more fluid). In any case, system fluidity

appears to be subordinate to system size in controlling shift duration, with the size versus time relationship remaining significant and positive across all five models.

**Mechanisms in real and model systems.** The empirical and modelled findings point to there being a fundamental mechanism in ecosystems that links physical size, structure and speed of failure. In terms of size and structure, the large body of ecological research on area-diversity relationships[35,36] allows us to assume that the self-organizational processes which create increasingly complex structures with, for example, more tropic levels, higher species richness or more sub-system modules, are strongly limited by space. It follows that large ecosystems will show disproportionately more complexity than small ecosystems. In that case, a recent analysis of coupled regime shifts[37] helps to identify two possible reasons why disproportionately high complexity in large ecosystems may instil resilience against a system reaching a regime shift, but once triggered provides favourable structures for failure: (1) there is a greater probability in larger systems that a 'shared' driver initiates synchronous failure in sub-system 'modules' at more than one location; and (2) there is a greater probability that the weak feedback mechanisms that maintain the stability of large, mature systems will be dominated by the emergence of stronger, 'hidden' feedbacks that progressively raise the probability that the failure of one sub-system will affect the stability of a neighbour. These two points are illustrated, respectively, by current concerns about the effects of disparate fires on the long-term resilience of the Amazon forest to climate change[25]; and the rapid spread of recent (2019-20) bush fires in SW Australia caused by existing fires igniting further fires[38].

The structural experiments support these ideas. From a network perspective, the LC model shows that systems with

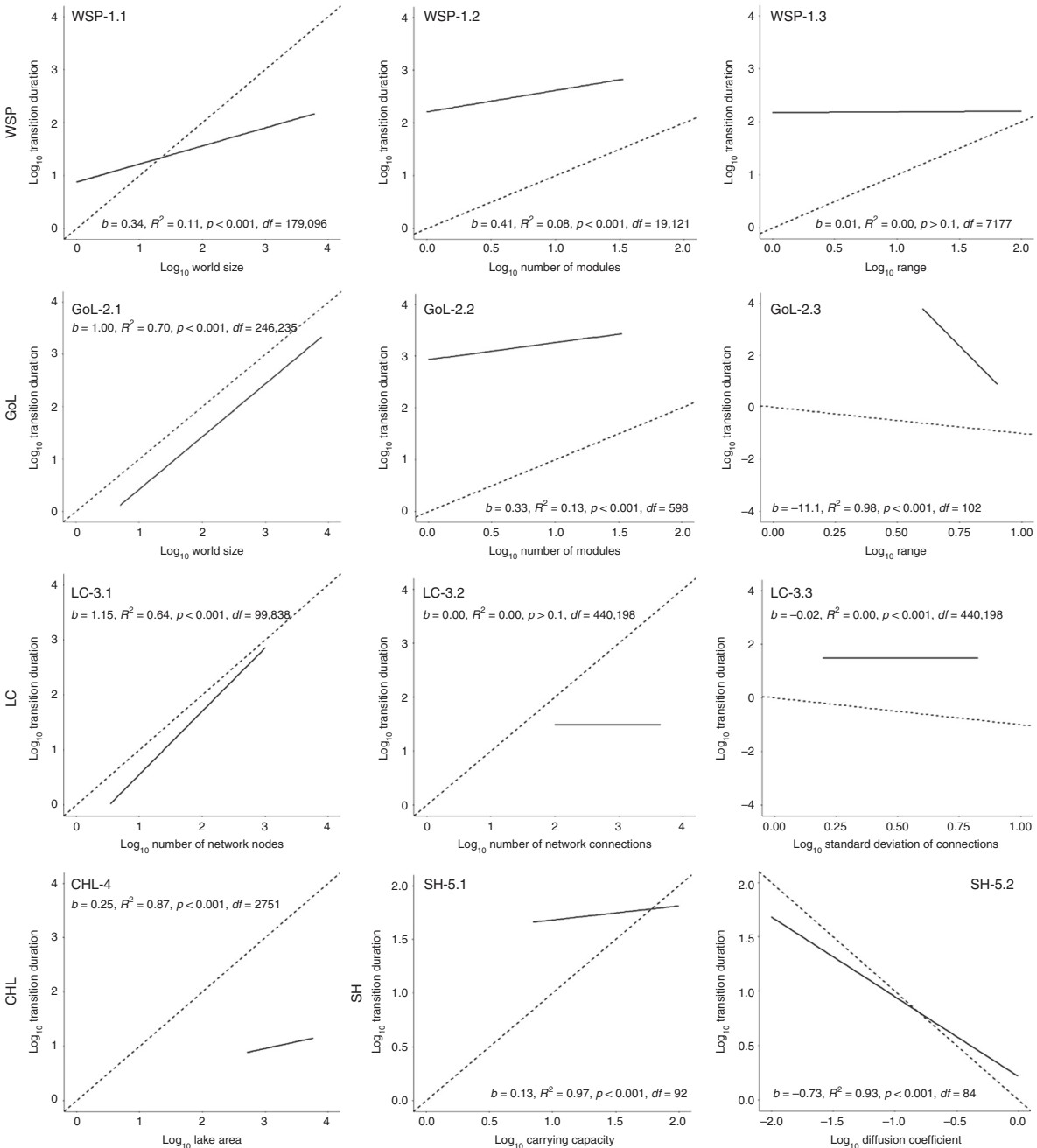

**Fig. 3 Modelled outputs exploring the relationships between regime shift duration and twelve spatial characteristics.** The trend lines and regression coefficients resulting from the twelve simulation experiments (#1.1–#5.2) show the effects of different spatial properties on the duration of system shift (Table 1). Dashed lines are 1:1 reference lines plotted with a *y*-intercept of '0'. Log-log axes are used for consistency with Fig. 1, with the 'b-term' representing the slope of the regression model. See Supplementary Table 8 for the linear model coefficients and comparisons.

heterogenous connections (Fig. 3) generally shift more rapidly than networks with relatively homogenous numbers of connections per node. This reflects the idea of 'keystone' nodes (Supplementary Fig. 8) which once flipped (e.g. 'black' to 'white') help to transmit stresses rapidly across the network, an interpretation that is consistent with the greater vulnerability of scale-free networks to a targeted attack on keystone nodes[19]. As expected, large systems of all kinds transmit stresses more slowly through greater distances, with some empirical (Supplementary Fig. 4) and modelled evidence suggesting fluid systems tend to transition more quickly (Fig. 3). However, this relationship is also sub-linear, implying a law of diminishing returns with each increase in system fluidity resulting in a disproportionately

smaller decrease in shift duration. Therefore, while the general negative association between fluidity and shift duration reflects the tendency for relatively resilient locations to emerge within system structures with lower connectiveness[16], more fluid systems (i.e. less modular) may lead to longer regime shift durations than would be otherwise expected under linear scaling. In other words, the ability of increasing fluidity to inhibit the rate of overall system transitioning may gradually weaken as the connectivity increases.

These findings on ecosystems invite a prediction about scale and shift duration in social and economic systems. It has been theorised that the difference between sub-linear and super-linear scaling with size is linked to the different controlling effects of

system structure in ecosystems versus system interactions in social systems[17]. This would imply that, in contrast to ecosystems, the collapse of social or economic systems (e.g. inter-bank trading) should scale super-linearly with disproportionately slower shift durations. Consistent with this, the empirical power-law relationship identified here becomes less sub-linear once social collapses are included (Supplementary Fig. 2); however, too few empirical data currently exist to fully and robustly explore this hypothesis.

**Implications for governance**. Ultimately, our findings have multiple implications for the governance of real-world systems. First, from local to sub-continental scales ($10^0$–$10^6$ km$^2$), we must prepare for regime shifts in any natural system to occur over the 'human' timescales of years and decades, rather than multigenerational timescales of centuries and millennia. Second, the apparent long-term stability of the largest, least disturbed ecosystems is a deceptive guide to the potential speed of their collapse. Therefore, the self-organising mechanisms (e.g. modularity) that help to instil systems with resilience prior to a tipping point may have limited ability to control the rate of collapse once a shift has been triggered. Third, homogenously connected systems shift relatively less quickly, meaning that ecosystems that are already disturbed but stabilised, or those that are engineered, may be relatively slower to collapse because of the lack of vulnerable modular structures. Thus, although shifts in agroecosystems are expected due to climate change[39], their relatively slow transitions may offer vital time for adaptation. Fourth, the 'window of opportunity'[15] open to divert unsustainable system trajectories is comparatively short for relatively small systems, meaning contingency plans should be formulated in advance and ready to implement across localised systems recognised to be heading towards the brink.

The exponentially increasing global trends of many social and biophysical variables over the past 65 years are widely viewed as unsustainable[40]. Along with the evidence for increasingly strong reinforcing feedbacks, interactions and couplings between variables[37,41,42], there is growing awareness around the heightened risk of current anthropogenic activities triggering sub-global regime shifts[14]. Combined with the findings presented here, humanity now needs to prepare for changes in ecosystems that are faster than we previously envisaged through our traditional linear view of the world, including across Earth's largest and most iconic ecosystems, and the social–ecological systems that they support.

## Methods

**Literature search strategy, case-study qualifications and dataset**. The literature search used three electronic databases, namely the University of Southampton's DelphiS interface, Web of Science and Google Scholar. Prospective case-studies were recognised by individual cases or combinations of the following key terms appearing within either the article title or abstract: regime shift, critical shifts, shift, abrupt shift, threshold change, tipping point, stark shift, abrupt change, human-natural system, ecosystem, ecological, social–ecological, ecosystem, irreversible, landscape, environment.

The literature search was carried out from February to August 2018. Date limits were not imposed on the year of publication. In addition, case-studies from both the Regime Shift Database[7] of the Stockholm Resilience Centre and the Threshold Database[20] of the Resilience Alliance were considered for inclusion. Each potential case-study, including the social systems included in Supplementary Fig. 2, had to then meet the following three-part criteria to be included in the empirical dataset of this study:

1. For inclusion based upon the characteristics of the regime shift, each case-study must exhibit:

   (a) A demonstrated/observed state change in a real-world environment, rather than just hypothesised or modelled.
   (b) Recognisable and clearly defined alternate states, consistent with common definitions, including both quantitative (e.g. ecosystem service availability) and qualitative (e.g. structural change) indicators.
   (c) Driver(s) of change that are beyond natural and/seasonal variations/ cycles.
   (d) Irreversibility over the temporal horizon of the original study.

   (e) Or, if reversed, human-led remediation efforts (e.g. artificially manipulating water quality) were completed over the course of the study.

2. In order to confidently and consistently measure the spatial extent (and depth) of shifts, the following steps were applied:

   (a) Use regime shift area (and depth where applicable) directly quoted in the case-study publication.
   (b) Ascertain whether the shift occurred across the whole system or sub-system of wider geographical entity, then:

      i. Consult 'Locational Information/Case Study/Methodology' sections of scientific publications to find quoted area of shift.
      ii. If shift occurred across entire system, we searched within related scientific publications to find extent (and depth) of system.
      iii. Widening the search to institutional literature, such as maritime management reports.

3. In order to confidently and consistently measure the temporal scale of shifts (i.e. the time taken to transition to a stable but functionally different system state), the literature either:

   (a) Directly quoted the shift duration in text.
   (b) Explicitly depicted shift duration in a time series of system conditions, with the significant deviation from the preceding regime flagged.
   (c) Visually estimated shift duration from a time series of system conditions. To remain consistent, the tipping point was always identified by the first sign of significant divergence from the preceding trend.

After applying the above qualifications, the final dataset (Supplementary Table 1) includes 42 regime shifts observed in nature (25 marine, 13 freshwater and 4 terrestrial).

**Sensitivity analysis of empirical results**. The empirical dataset suggests that there is an overarching positive association between system area and shift duration, and that larger systems tend to shift comparatively quickly relative to their size. However, it is reasonable to ask questions around the uncertainty of this result. Therefore, we investigate the extent to which the sub-linear trend is (i) dependent on any one datapoint in the empirical dataset, and (ii) affected by uncertainties within the empirical dataset. Regarding point (i), we created 42 new empirical datasets, each with one of the empirical records removed. We fitted power-law relationships to each of the new 42 datasets (each with 41 empirical records) and assessed the degree to which removing any one empirical record impacted the production of a significant, sub-linear association between system area and shift duration. We undertook a simple Monte Carlo analysis to investigate point (ii). For each of the 42 empirical records, 5000 random error terms were generated, converting the shift durations to values between 50 and 150% of their original values. The resulting error ranges are graphically represented in Supplementary Fig. 5. Error terms were only applied to the shift durations, as confidence in the system area values is relatively high (Supplementary Table 1). From here, we fitted power-law regression models through each of the 5000 new models and recorded the resulting slope and significance coefficients. All analyses were conducted using the statistical software R[43].

**Model selection strategy**. The model search was carried out from February 2018 to February 2019, during which we identified models that reflected the characteristics of the empirical regime shifts data obtained.

1. For inclusion based upon the characteristics of the regime shift, each model must exhibit:

   (a) A state change.
   (b) Recognisable and clearly defined alternate states, consistent with common definitions, including both quantitative (e.g. ecosystem service availability) and qualitative (e.g. structural change) indicators.
   (c) Variables acting as driver(s) of change.

2. In order to confidently and consistently measure the temporal scale of shifts, the model either:

   (a) Explicitly depicted shift duration in a time series of system conditions, starting in an unstable state, where the start of the shift is assumed to be the start of the model run.
   (b) Started in a stable state, from which shift duration could be estimated from a time series of system conditions. To remain consistent, the tipping point was always identified by the first sign of significant divergence from the preceding trend (see the 'Identifying regime shift durations of modelled time series' section for more details below).

3. In order to investigate the impact of system characteristics, the model either:

(a) Allowed for variation in system size.
(b) Allowed for variation in metrics of system fluidity or connectedness.

4. Finally, in accordance with FAIR principles[44], models were required to be open-access.

After applying the above qualifications, we obtained five models of regime shifts that are findable, accessible and reusable as well as being comparable to our empirical data. Of these models, two are known to illustrate tipping points and hysteresis (CHL and SH models). The models are described below.

**The Wolf-Sheep Predation (WSP) agent-based model**. The WSP model explores the stability of predator-prey relationships[21]. The construction of this model is described in two principle articles[45,46]. In our investigation, we used a variation of the model which includes grass in addition to wolves and sheep. Both wolves and sheep are randomly generated and move randomly through a landscape. Each step costs both animals in terms of energy; wolves must eat sheep and sheep must eat grass in order to replenish their energy. Therefore, any animals that run out of energy die. Once grass has been eaten, it will regrow after a fixed number of model steps. Finally, every animal has a fixed probability of reproducing at each time step. This model is freely available within the NetLogo software[47] and the default values for the model variables are shown in Supplementary Table 3. The WSP model outlined above is sometimes stable[21], but can be made unstable by varying the grass regrowth time. Once the model is unstable, it can be observed to go through three possible regime shifts (Supplementary Fig. 7): (1) the extinction of wolves, (2) the extinction of sheep, (3) the progression of the landscape to full grassland, which with no grazers present, could lead to succession towards another ecosystem state. By altering specific variables and then destabilising the WSP model we were able to investigate the impact of those variables on the duration of the regime shifts. The variables we investigated using the WSP model were system area, module size, and system fluidity (Table 1). To investigate the impact of the area of the landscape on the duration of the regime shift, we increased the length and width of the landscape by two pixels at a time between 1 and 100, while maintaining constant starting densities of both wolves and sheep (Supplementary Table 3). To ensure unstable systems, the reproduction rates of sheep were altered to a constant of 7% and grass regrowth time was varied from 1 to 100. Using the 'BehaviorSpace' tool within Netlogo[47] every variation of this model was run for 5000 time steps, unless all three regime shifts occurred prior to this. This process resulted in 260,100 model runs. To investigate the impact of the size of modules within the landscape on the duration of the regime shift, we varied the height of the landscape between 2, 5, 10, 20, 50, and 100 cells. Here we again maintained constant starting densities of both wolves and sheep, but summed model runs together so that world size was consistently 100 × 100 pixels (Supplementary Table 4). To ensure unstable systems, the reproduction rates of sheep were altered to a constant of 7% and grass regrowth time was varied from 1 to 100. As per the world size experiment, every variation of this model was run for 5000 time steps, unless all three regime shifts occurred prior to this. This process was repeated 100 times, resulting in 930,000 model runs. To investigate the impact of the system fluidity on the duration of the regime shift, we varied the mobility of the animals between 1 and 100, while maintaining a constant landscape size of 100 × 100 pixels (Supplementary Table 4). To ensure unstable systems, the reproduction rates of sheep were altered to a constant of 7% and grass regrowth time was varied from 1 to 100. Every variation of this model was run for 5000 time steps, unless all three regime shifts occurred prior to this. This process resulted in 10,000 model runs. The GoL model can be obtained from the following URL: https://ccl.northwestern.edu/netlogo/models/WolfSheepPredation

**Game of Life (GoL) cellular automaton model**. In the two-dimensional GoL each cell can be either one of two possible states: 'alive' or 'dead'. At every time step, each cell checks the state of itself and its neighbours, and then sets itself as either alive or dead based on its neighbours' states. This model is freely available within the NetLogo software[22] and the default values for the model variables are shown in Table S5. The GoL model outlined above is inherently unstable when an initial density of 35% is used (i.e. the system begins to shift from the initial state to the alternative state as soon as the model run begins). Upon starting the model at this density, the number of 'alive' cells decreases until a stable state is reached. Thus, the system can only be observed to go through one possible regime shift: from an unstable state with both alive and dead cells to an alternate stable state in which either all cells are dead, or a stable mixed state has been reached. By altering specific variables we were able to investigate the impact of those variables on the duration of the regime shift, starting from an unstable state. The variables we investigated using the GoL were system size, module size, and system fluidity (Table 1). In order to determine when stability occurred, we inserted a new stop function (Supplementary Note 3) which would stop the model if the number of living cells did not change for 100 time steps. To investigate the impact of the size of the landscape on the duration of the regime shift, we increased the length and width of the landscape by two pixels at a time between 1 and 100, while maintaining consistent densities of both alive and dead cells (Supplementary Table 5). To ensure unstable systems, the initial density was set to 35%. Using the 'BehaviorSpace' tool within Netlogo, every variation of this model was run for 5000 time steps, unless a stable

state was reached prior to this. This process resulted in 260,100 model runs. To investigate the impact of the size of modules within the landscape on the duration of the regime shift, we varied the height of the landscape between 2, 5, 10, 20, 50, and 100, again while maintaining consistent starting densities of both alive and dead cells, but summed model runs together so that word size was consistently 100 × 100 pixels (Supplementary Table 5). To ensure unstable systems the initial density was set to 35%. Every variation of this model was run for 5000 time steps, unless a stable state was reached prior to this. This process was repeated 100 times, resulting in 930,000 model runs. To investigate the impact of the system fluidity of the landscape on the duration of the regime shift, we varied the number of neighbours each cell considered between 4 (i.e. von Neumann neighbourhood[34]) and 8 (i.e. Moore neighbourhood[34]), while maintaining a constant landscape size of 100 × 100 pixels and a constant proportion for the decisions to 'die' (Supplementary Table 5). To do this, we further adapted the standard GoL code, updating the 'to go' function to include both possible neighbour combinations (Supplementary Note 3). To ensure unstable systems, the initial density was set to 35%. Every variation of this model was run for 5000 time steps, unless a stable state was reached prior to this. This process was repeated 100 times, resulting in 200 model runs. The GoL model can be obtained from the following URL: https://ccl.northwestern.edu/netlogo/models/Life

**Language Change (LC) network model**. The LC model explores how the structure of social networks can affect the course of language change[23,24] (Supplementary Fig. 8). In our investigation, we used a variation of the model (termed 'individual') in which individuals can only access one language at a time. Each time step, individuals choose one of their neighbours randomly and then adopt that neighbour's language (Language 1 or Language 2). This model is freely available within the NetLogo software[47] and the default values for the model variables are shown in Supplementary Table 6. The LC model outlined above is inherently unstable. Language 1 is created as dominant and cannot be lost once adopted[23]. Thus, the system can only be observed to go through one possible regime shift: from a mixed state with two languages to an alternate state whereby language 1 has become saturated in the population (Supplementary Fig. 8). By altering specific variables, we were able to investigate the impact of those variables on the duration of the regime shift, starting from an unstable state. The variables we investigated using the LC model were number of connections, number of nodes and network connection heterogeneity (Table 1). To investigate the impact of the number of connections in a network on the duration of the regime shift, we varied the number of connections between 99 and 4500 (Supplementary Table 6). In order to do this, the number of connections was added as a global variable and the code to create the network was altered to ensure the number of connections between nodes was equal to this user-defined value (Supplementary Note 4). Using the 'BehaviorSpace' tool within Netlogo, every variation of this model was run for 5000 time steps, unless the regime shift occurred prior to this. This process was repeated 100 times, totalling 440,200 model runs. To investigate the impact of the number of nodes in a network on the duration of the regime shift, we varied the number of nodes between 3 and 1000 (Supplementary Table 6). The number of connections was set to one but would default to the number of nodes minus one to ensure all nodes were connected. Every variation of this model was run for 5000 time steps, unless the regime shift occurred prior to this. This process was repeated 100 times, totalling 99,800 model runs. Instead of re-running the LC model to specifically investigate the impact of network connection heterogeneity on the duration of regime shifts, we maximised computational efficiency by analysing network heterogeneity in the systems used to investigate the number of connections. During the above LC model experiments, we recorded the standard deviation of the number of connections of each link; acting as an appropriate measure of the heterogeneity of the connection distributions as the underlying distribution is normal (Gaussian; Fig. S8). The LC model can be obtained from the following URL: https://ccl.northwestern.edu/netlogo/models/LanguageChange

**Lake Chilika fishery (CHL) system dynamics model**. The CHL model[25] was built to investigate the future social–ecological sustainability of the Chilika lagoon—Asia's largest brackish water ecosystem—located in Odisha, India. Essentially, the model simulates the coupled effects of various biophysical and socioeconomic pressures on the fish stock. As a system dynamics model, the key dynamics of the social–ecological system are represented as stocks (e.g. fish population, lake water sediment and aquatic vegetation), flows (e.g. freshwater and climatic inputs, fish births and deaths) and feedbacks (e.g. fishery intensification) which all evolve over time. Each model time step equals 1 month, although outputs are generally aggregated to the annual resolution to improve visualisation. Here, simulations are run for 1524 time steps, equalling 127 years (i.e. the period from 1973–2099). In the original model[25], the model simulates four socioeconomic stresses on the fish population: (i) the number of fishers able to generate their livelihood from the fishery is related to a simple carrying capacity, based on the economic revenue of the fish catch, the average income of each fleet (i.e. traditional and motorised) and the minimum cost to fish; (ii) relatively affluent traditional fishers may switch from traditional wind-assisted sailing boats to relatively fish catch intensive motorboats; (iii) the number of days fished each month is proportional to the underlying density of the fish population; (iv) while the acceptance of juvenile catch increases during stock declines to compensate for lost fishing days. The original model also

captures three biophysical pressures on the fish population: (i) the effect of tidal outlet sedimentation and closure on the migration of fish to and from the Bay of Bengal, with 70% of the fish stock undertaking this migration pathway each year to complete their natural breeding cycles; (ii) the effects of lake water salinity, temperature and dissolved oxygen concentration on the survival rate of juvenile fish per unit time; (iii) the growth of surface water aquatic vegetation which provides refuge from fishery activities. The model also simulates the effects of alternative governance options, including the implementation of fishing bans and the frequency of tidal outlet maintenance (i.e. removal of accumulated sediment). The model is aspatial, as per the majority of system dynamics models. However, the model does simulate the effect of lake area on the growth of aquatic vegetation and the volume of rainfall falling directly onto the lake—with subsequent impacts on the salinity of the lake water and the accumulation of lacustrine sediment (i.e. larger area leads to higher direct rainfall inputs, leading to greater flushing of sediment from the lagoon). Therefore, to model the direct association between lake area and the duration of transition, we turn off all socioeconomic pressures (i.e. set fish catch from both fleets equal to zero). In turn, we vary the parameter named 'Chilika area $km^2$' between 500 $km^2$ and 10,000 $km^2$ (i.e. 50–1000% of the original lake value). The model is run for 5000 simulations in the sensitivity analysis mode, sampling a different lake area between the minimum and maximum area limits per simulation. Tidal outlet maintenance is turned off, meaning the lacustrine sediment is allowed to accumulate naturally. Similar to the WSP model (Supplementary Table 3), the model may remain stable across the simulation horizon. Therefore, the breakpoint function[48] is used to detect the onset of the shift, and the end of the shift is flagged once the fish population falls beneath 1% of the fish population recorded at the start of the simulation (Supplementary Fig. 10). The model exhibits fold bifurcation behaviour and hysteresis; for example, in Supplementary Fig. 11, whereby attempts to recover the collapsed fish population require the stressor (i.e. lake salinity, which is a proxy for lake sedimentation and tidal outlet closure) to be reversed back past the point that caused the original transition. The model is available on reasonable request from the authors of the original study[25].

**Spatial Heterogeneity (SH) model**. The SH model is an illustration of how spatial structure can affect the potential of systems to oscillate, particularly how stabilization can arise through spatial heterogeneity[2]. The SH model is known to show Hopf bifurcations[2]. The model uses predator-prey relationships to represent the interaction between zooplankton and algae co-existing within a lake but simplifies the spatial processes by considering zooplankton to be situated in one part of the lake, while algae are present throughout (Supplementary Fig. 12). Thus, in one compartment (A1) zooplankton graze the local population of algae, but the algae within the other compartment (A2) are predation free. The model experiments observe the shift to a state where the zooplankton are extirpated (Supplementary Fig. 13). Reference #2 provides a detailed description of the model's original rationale and application. Here we show the reproducible Netlogo code (Supplementary Note 5) and the model parameters (Supplementary Table 7). To investigate the impact of the size of the ecosystem on the duration of the regime shift, we increased the carrying capacity of algae (K) by one, varying from 1 to 100 while maintaining constant parameters for all other variables (Supplementary Table 7). Using the 'BehaviorSpace' tool within Netlogo[47], every variation of this model was run for 10,000 time steps, resulting in 100 model runs. To investigate the impact of the system fluidity on the duration of the regime shift, we varied the fraction of volume exchanged between inside and outside (d; Supplementary Fig. 12) between 0 and 1 in increments of 0.01 (maintaining all other variables as constant; Supplementary Table 7). Every variation of this model was run for 10,000 time steps. This process resulted in 101 model runs. The SH model can be obtained from ref. [2].

**Identifying regime shift durations of modelled time series**. The completed model runs detailed above were exported as comma-separated values and read into the statistical software R[43] for analyses. The University of Southampton supercomputer 'Iridis 4' was used to process the model outputs. To demark the start of the regime shifts for the WSP model that starts stable (Supplementary Fig. 6), we used the breakpoints function within the R-package 'strucchange'[48]. The breakpoint function is based upon finding significant deviations from stability in classical regression models, whereby the regression coefficients shift from one stable regime to another[48]. We assume a priori that the number of statistically distinct time-series segments is equal to two: (1) pre-collapse state, (2) collapsed state, for the wolves, sheep and grass trends. Therefore, the breakpoint function searches for a single optimal breakpoint for each trend. Then for wolves and sheep, the end of the shift occurs once their respective abundances equal zero (Supplementary Fig. 7a,b), while the termination of the grass shift occurs once grass completely covers the system (Supplementary Fig. 7c). As detailed above, the Chilika model uses the same breakpoint strategy as the WSP model, with the breakpoint function detecting the shift from the first stable regime, and the end of the shift denoted by the first time step that the fish population is less than 1% of the original fish population. The breakpoint function is not required for the LC, GoL or SH model runs, as the models starts in an unstable state and so the start of the regime shift coincides with the first time step of the model. Therefore, in the LC model, the shift duration is equal to the number of time steps (from start) until all the system nodes have the same language state (Supplementary Fig. 8). In the GoL model, the shift duration is

equal to the number of time steps (from start) until the model reaches a stable state in which either all cells are dead, or a stable mixed state has been maintained for 100 steps (Supplementary Fig. 9). Likewise, in the SH model, the shift duration is equal to the number of time steps (from start) until the first time step when the concentration of zooplankton equals zero (Supplementary Fig. 13). To produce the regression models from the modelled data (Fig. 3), the model runs that did not undergo shifts (as explained in this section) were omitted from analysis (Supplementary Table 9). Then, the log–log linear models were formulated, relating shift time to the variable of interest (Table 1). Variations in the rates of grass regrowth were accounted for within the WSP generalised linear models to assess the effect of the independent variable (Table 1) on shift duration for a given disturbance rate.

**Reporting summary**. Further information on research design is available in the Nature Research Reporting Summary linked to this article.

## Data availability
All data generated or analysed during this study are included in this published article (and its supplementary information files).

## Code availability
All model code is freely available from the following citation numbers in the reference list below: 2 and 21–25. The complete Lake Chilika fishery model can be obtained from the corresponding author. The code amendments used to produce results presented in this paper are detailed in Supplementary Notes 3–5.

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

## Acknowledgements
G.S.C. and J.A.D. gratefully acknowledge a research studentship and financial support respectively from the Deltas, Vulnerability and Climate Change: Migration and Adaptation (DECCMA) project under the Collaborative Adaptation Research Initiative in Africa and Asia (CARIAA) program with financial support from the UK Government's Department for International Development (DFID) and the International Development Research Centre (IDRC), Canada (Grant No. 107642-001). The views expressed in this work are those of the creators and do not necessarily represent those of DFID and IDRC or its Boards of Governors. S.W. was funded by UKRI project numbers: NE/L001322/1, NE/T00391X/1, ES/R009279/1 and ES/R006865/1. We thank Professor Peter Langdon (University of Southampton) for providing early contributions to the concepts of this paper and Dr Rong Wang (Nanjing Institute of Geography and Limnology) for preliminary data collection and analysis. We also thank Dr James Dyke (University of Southampton), Professor Felix Eigenbrod (University of Southampton) and Dr Robert Cooke (University of Southampton) for their comments on earlier versions of the paper. We also wish to acknowledge the use of the IRIDIS High Performance Computing Facility, and associated support services at the University of Southampton, in the completion of this work.

## Author contributions
G.S.C. contributed towards the empirical data collection, analysis of model outputs and manuscript writing. S.W. contributed towards the study design, computational modelling, analysis of model outputs and manuscript writing. J.A.D. contributed towards the study design, theoretical development and manuscript writing.

## Competing interests
The authors declare no competing interests.
