## [Peer Review File · Nature Communications]

Reviewers' Comments:

Reviewer #1:

Remarks to the Author:

General comments

The manuscript "Regime shifts occur disproportionately faster in larger ecosystems" poses an interesting question about the role of scale and speed in regime shifts. The key finding is a sub-linear power law relationship between system size and duration of regime shifts. The work is compelling and the results relatively well supported by case studies from literature review (when scales of shifts has been possible to identify) and a modelling exercise that performs a sensitivity analysis on system size, modularity and fluidity. I believe the paper is well suited for the audience of the journal. However, at its present state it requires some work on clarifications to be ready for publication.

My main concern is about how duration is treated both in the empirical cases and the models compared. The authors seems to mix in different places of the paper slightly different meanings of duration. Sometimes the reader is guided to understand duration as the amount of time between crossing a tipping point —the onset of the shift— and the realisation of an alternative regime. In other parts of the paper duration is treated as the amount of time spend in regime one before shifting to the second regime. This second interpretation is more prevalent when using agent based models where the breakpoint function were not used.

My recommendation here is to clarify what is meant by duration through the paper and be consistent with the definition. What was used for example to construct the empirical basis of your study (Table S1): (i) duration of regime one before shifting to the alternative regime, (ii) time between crossing a threshold and realising the alternative regime, or (iii) time spend in the alternative regime before returning to regime one? — Did you extracted the time series from each paper and run the breakpoint function? Having that clear would help the reader to better interpret your key results.

To support clarity, I'd suggest to add one or two models to your modelling section in the paper where regime shifts are specifically addressed. Currently, the WSP model is the only one that lends itself for testing with the breakpoint function. I'd suggest to add one or two models that address specifically ecological regime shifts, e.g. forest to savanna and eutrophication. Many of these models already exist, see for example Scheffer's book on Critical Transitions in Nature and Society (2009), or Solé book on Phase Transitions (2011) — both have useful appendixes with various models, some of which are spatially explicit. I believe the WSP is the only model with a first order critical transition, while the GoL and LC seems to be second order critical transitions: they are continuous in the first derivative but discontinuous in the second. That's why you are "forced" to use time spend in regime one as proxy of duration, but most importantly for their implications on real regime shifts, GoL and LC does not seem to have hysteresis (or correct me if I'm wrong). The motivation for two extra models is to allow yourself to test the shift duration as the time from crossing the threshold to realising the alternative regime, which is what the breakpoint function tries to get. Additionally, having a in-built threshold on a slow variable also allows you to test duration in the case of hysteresis, when the path to recovery differs from the path to collapse in the bifurcation. Is the duration the same when hysteresis is at play?

I would also like to encourage you to interpret your key finding. A power law is a statistical pattern, but it would be very useful for the reader (and the impact of your paper) if you can provide an informed speculation of why that pattern emerge? What do you think is underlying that 0.8 slope on the log-log scale? While I was reading your paper I was also reading the book Scale by Geoffrey West, who finds many power law distributions in a number of different systems —they are very important

features of complex systems— but also guides the reader on why they emerge, what do we know so far about their generative mechanisms or what are we missing. I believe a few sentences on the question why, even if speculative, can inform future hypothesis to explain that pattern.

Below you will find specific comments. I hope you find them useful.

Best regards,

Juan Rocha

Specific comments

Line Comment

51-52 I did not understand “why in reverse”. Could you expand a bit?

98 It is hard to conceptualise clear regimes in the GoL, perhaps some live cells vs all death. But then it is irreversible and does not have hysteresis. How these models compare with the cases reported in your empirical analysis? They seem to be reversible, at least some of them on human time scales.

127 Doesn't this contradict the network finding. GoL with 4 or 8 neighbours is the same as a homogeneous network with degree 4 or 8.

132-34 This is a bit of an overstatement — if I understood correctly the operationalisation of fluidity in the modelling exercise, it is defined as how far a disturbance spreads. You are assuming that the disturbance in water is more fluid than in land. But that is not necessarily the case. Climate change impact both at large scales, but other more localized perturbations might follow similar patterns of disturbance in land and sea i.e fishing and deforestation — both require mobility and humans are constrained by place and tech to move.

143 “...systems of all kinds transmits stresses more slowly through greater distances and absolute modularity” — again, it depends of the disturbance. Another aspect not discussed in the paper is the time scale of the species involved. In marine systems primary produces are often plankton (generation time of weeks), while terrestrial systems are slow growing species (trees generation times of centuries). Thus it might not have to do with the duality terrestrial vs. marine, but rather the time scales of the species involved.

151 Be careful with this statement: the existence of key nodes or influential nodes depends on network structure. For the purpose of disintegrating the network, targeting hubs is effective in scale-free networks. For the purpose of changing the state of notes, hubs are not necessarily the best and it also depends if the speeding process underlies a simple or complex contagion dynamic. For small-world networks, it depends where the network is in the spectrum of high-low clustering, not all small-world networks can be controlled with high degree nodes.

Supplementary information:

38 Does this include time spend on regime 1 or time since tipping point crossed? Since the empirical data does not necessarily have information about thresholds and when they were crossed, how was the duration measured? Did you used the same breakpoint function on the time series of the papers selected?

103 I didn't understood well enough the size of modules experiment. I'm assuming a module is the height of the landscape, and then the sum is necessary for comparison purposes, but also means that there is series of say 2-cells landscapes glue together but that do not influence each other. Is that

right? If they do influence each other, how is the experiment different from the size of landscape — I guess what is not clear to me is what is a module on the GoL. I understand a module as a unit of a landscape or a network with more interactions than with its surroundings, but it is still connected to its surroundings.

149-51 Why is it the standard deviation a good measure of heterogeneity? If the underlying distribution of degree is normal (Gaussian) then it's a good approximation, but for other degree distributions such as small-world or scale-free, it can be misleading. In heterogeneous networks there is a negative correlation between in- and out-degree. A common measure is $\langle |k_{in} - k_{out}| \rangle / \langle k \rangle$ which lands in the range 0-1 and is easy to interpret.

169-70 There is no Table 2 in the paper (results from a GLM?), and when omitting observations in a regression, it's a good practice to state how many observations were omitted.

Reviewer #2:

Remarks to the Author:

In the submitted manuscript the author(s) examined the empirical relationships between size and transition duration of systems that experienced regime shifts and used three different simple models to demonstrate the role of system fluidity, connectivity and modularity in causing such relationships. The paper is clearly written. I find this work very interesting and highly relevant to the hot topics critical transitions and ecosystem resilience. To my knowledge this relationship that was examined in this study has not been explicitly explored, presenting clear novelty. However, I have major doubts about the models, and feel that there are some concepts to be further clarified and additional analyses needed.

1. The author(s) focused on 'regime shifts' or 'tipping points' or 'critical transitions' in (socio-)ecological systems. These system behaviors of abrupt changes have been mostly linked with alternative stable states (ASS) whose mathematical underpinning is fold bifurcation. Indeed, the empirical database used in this study contains a range of systems that have been suggested to have ASS (and fold bifurcation). For any real-world system it is difficult to give solid proof that ASS underlies system behavior, however, in the modelled system this is doable in an explicit way. My major concern is that none of the simple models have been shown to have ASS. And in my opinion most (if not all) models would not have ASS as the essential element of positive feedback is largely lacking. I am well aware that one can argue that real-world regime shifts are not necessarily underpinned by ASS, and there are some other types of bifurcations other than fold bifurcation that may be the mechanism of the observed abrupt changes. Nonetheless, there is a clear mismatch between the real-world and modelled system behaviors in the sense of ASS vs. non-ASS. I think it is critical to include a couple of simple models that have been extensively used for the study of ASS such as the classical May's grazing model and some more recent models used for unravelling the collapse of pollination networks or social networks. I understand that would mean a lot of additional analyses, but the current modelling results are not really convincing.

2. For systems that have bifurcation behaviors, it would be great if the author(s) could systematically analyze the transient states if they simply focus on the duration of transition. But I feel it's also valuable if they could also perform systematic analysis on how systems size affects resilience because the duration of transition is also influenced by system resilience (depending how you define resilience) other than transient states when facing complex perturbation.

3. I would like to see analysis on the empirical relationship between size and duration of transition for different types of systems separately. It looks that they would have pretty different slope?

4. While the observed relationship between size and duration of transition is pretty clear, important caveats should be mentioned when it comes to inferring real-world situations by extrapolating the relationships, as they are plotted on a log scale, the predicted duration of transition could have a very

large range (as also pointed out by the authors). It brings large uncertainties for interpreting and predicting real-world transitions.

Reviewer #3:

Remarks to the Author:

This paper addresses an interesting question, which is: how does the speed of ecosystem collapse vary with system size? The paper uses information from 42 different ecosystems (assembled for existing data bases) to address that question. They show that the time to collapse increases with system size. So, larger systems collapse slower than smaller one, but the authors also show that they do it disproportionately faster than smaller systems. The authors then use 3 models to investigate the role of 'system fluidity', 'connectivity' and 'spatial modularity' in driving the speed of system's collapse: a wolf-sheep predation agent-based model, a game of life cellular automaton, and a language change agent-based model.

The data presented is interesting, but the authors don't really highlight the practical relevance of their main findings about the dependency of speed of collapse with size. The fact that system size influences ecosystem collapse in expected. How does the nonlinearity of the relationship change our understanding and predictive ability of ecosystem resilience to perturbations? Their results are also insufficiently supported by robust statistic (in particular whether the trend of that relationship linear or nonlinear). I am also not convinced by the modeling part: the choice of these minimal models is not justified and seems odd since none of them is related to the type of data analyzed; the model analysis is also not thorough enough to really be convincing.

I here below list some major concerns and some more minor comments on the manuscript and the supplementary information.

Major concerns:

1. The relevance of the main result is not sufficiently discussed. Larger systems shifts more slowly than smaller systems but disproportionately faster. The first part is very much expected. The second part may be more interesting, but:
 - how robust is that trend? Based on the figures and on the numbers in the text, the differences seems minimal. Unless I missed it, the authors did not show a statistical test of the comparison between the different fits.
 - The authors don't mention anything related to the uncertainty behind the data points, but based on the proximity of the two fits, couldn't it be that this result becomes not significant when taking into account all the uncertainties behind the data?
 - Even if the difference between the two fits is statistically significant, and the nonlinear fit is indeed better, how relevant is that difference (in magnitude)? The estimates of time to shift seem to be really close in both cases, and again could well be within the range of uncertainty.
 - What are concretely the implications of such results? Why should this result affect the way we look at these ecosystems? We live in an increasingly fragmented world; what do these results mean in that context?
2. The data is insufficiently presented and discussed. It seems to assemble very different studies, conducted by different people, at different scales, with different methodologies, for different purposes. What's the error on the estimation for times to shifts? Maybe this does not matter too much since the trends exists across orders of magnitude, but it would still be important to mention and discuss this.

3. I am puzzled by the choice of the models. These models really don't seem the best choice to test the author's idea. Why choosing these models? In what way are they complementary? Why not using actual ecosystem models and maybe also socio-ecological models? Also the presentation of the models lacked information about the types of behaviors that the models exhibit: What are the dynamical behaviors of these models? How do they behave along gradients? What makes them shift? Some of this information is provided in supplementary information, which is well written and clear. But some of the basic information would need to be moved to the main text for the reader to be able to understand the results. For example, I did not understand what the authors meant by 'module' and how this was implemented concretely in the simulations.

4. This is only a suggestion, but the results that I found the most interesting were the actual quantitative values given for times to shift in various ecosystems (l. 13-15). Maybe a more convincing way of presenting the results of this analysis could be to first mention that result (i.e. the fact that across ecosystem types, it seems that ecosystems are likely to collapse at years to decades timescales, meaning much faster than expected), then mention the dependency of time to collapse to system size, and finally discuss the problem of habitat destruction/fragmentation in that context?

Minor comments

l. 5-6: Is the frequency of regime shifts predicted to increase? Are there any formal study really demonstrating this or is this just a hypothesis based on a verbal argument?

l. 58:65: Move this part later in the text? This is a bit too detailed at this moment in the text. It would go better around l. 80.

l. 70-71: 'the overarching relationship is sub-linear': I guess that you tested that this fit was significantly better than a linear relationship? I might have missed it but this information does not seem to be clearly mentioned in the text and the statistics for the comparison of the fits are not shown (or I did not find them). Even if that's the case and a sublinear fit is indeed better than a linear fit, it's not really clear that this makes a significant difference for the estimation of transition time (see Fig 1 – I realize that the scale is log-log). It seems on the contrary, that it does not matter much except for very small systems? This should more thoroughly explained and discussed.

l. 83: It is not clear at this point of the text what those terms mean. Moving the text from l. 58-65 here might help. We also need more detailed information: what parameter of the model were varied concretely?

Table 1 is very difficult to follow. The headers of the first row should be changed to more explicitly describe their content.

Col 1: should mention the type of model (mathematical formalism: ODE, PDE, CA...)

Col 2: 'Experiment' → 'parameter varied'?

Col 3: very redundant with col 2: merge?

Col. 5: 'repeat' → replicas? Repetitions?

Col 4: 'runs' → 'simulation length'? 'number of timesteps'? Or is this the number of runs? In any case, there is not enough information in this table for us to understand what this means.

Actually I would consider moving this whole table to appendix.

l. 90: What does 'dynamic spatially distributed system' mean concretely? What kind of mathematical formalism does that correspond to?

l. 96-97: Are these alternative stable states?

Figure 2 is difficult to understand. It would really help to have the names of columns (use the same name as in the text and table 1 for the 3 parameters varied) and rows (i.e. names of models). Along the x-axis: it would help to use the same terminology as in the text. For at least 6 of the 9 panels, it's very unclear whether linear or non-linear fit are better. It would be nice to show the outcomes of a statistical test.

I. 144-145: Is that seen in the data?

I. 156: What is 'self-organisational energetics'?

I. 173 and following: These estimates are interesting, but how do you take modularity and connectivity of the ecosystems into accounts? Do you suppose these whole ecosystems are one homogeneous fully connected system?

L. 186-187: This sentence seems to contradict the network literature on modularity: modules are usually thought to confine the spread of perturbations in complex systems, and they are therefore usually found to be stabilizing (which I would expect should lead to longer shift duration). As mentioned, I had a hard time understanding what was meant by 'modules' in this study, isn't it rather habitat fragmentation? (it's not really that the structure is modular but more that it is patchy)

I. 206: 'heightened risk of current activities triggering sub-global tipping points': what does that mean? And what is the evidence for that? This paper does not seem to provide elements to support such a claim.

I. 207: 'with the findings presented here, humanity now needs to prepare for even faster change in shifting ecosystems': again, I don't see how this is in agreement with the results presented here. In what sense do the results suggest an acceleration of shifts?

In the SI, for the Wolf sheep predation model, I did not understand what was meant for the modules. The SI mentions (l. 70 of SI) changing the height of the landscape but does not that correspond to changing the size of the landscape?

Again, in the SI, regarding the Game of life model (l. 88 of the SI), what does it mean that the model is 'inherently unstable' when cover is below 35%? That the system goes to global extinction? L. 90-91 seems to contradict that. That there are cyclic dynamics? Again for this model, I did not understand the meaning of modules.

NCOMMS-19-10545A – Responses to referees' comments

Reviewers' comments:

Reviewer #1 (Remarks to the Author):

General comments

1. The manuscript “Regime shifts occur disproportionately faster in larger ecosystems” poses an interesting question about the role of scale and speed in regime shifts. The key finding is a sub-linear power law relationship between system size and duration of regime shifts. The work is compelling and the results relatively well supported by case studies from literature review (when scales of shifts has been possible to identify) and a modelling exercise that performs a sensitivity analysis on system size, modularity and fluidity. I believe the paper is well suited for the audience of the journal. However, at its present state it requires some work on clarifications to be ready for publication.

My main concern is about how duration is treated both in the empirical cases and the models compared. The authors seems to mix in different places of the paper slightly different meanings of duration. Sometimes the reader is guided to understand duration as the amount of time between crossing a tipping point - the onset of the shift - and the realisation of an alternative regime. In other parts of the paper duration is treated as the amount of time spend in regime one before shifting to the second regime. This second interpretation is more prevalent when using agent based models where the breakpoint function were not used.

My recommendation here is to clarify what is meant by duration through the paper and be consistent with the definition. What was used for example to construct the empirical basis of your study (Table S1): (i) duration of regime one before shifting to the alternative regime, (ii) time between crossing a threshold and realising the alternative regime, or (iii) time spend in the alternative regime before returning to regime one? - Did you extracted the time series from each paper and run the breakpoint function? Having that clear would help the reader to better interpret your key results.

We have made a number of edits to clarify our definition of regime shift duration and address the reviewer's concerns:

- We have added additional text to better clarify our conceptual understanding of regime shifts and the measurement of their duration (*lines 32-35*)
- We have also added extra text to the Methods section, namely *lines 308-311* and *lines 500-511*
- We have added three graphs (*Fig. S9, Fig. S10* and *Fig. S13*) that explicitly visualise our definitions of regime shift duration. Along with *Fig. S7*, these cover all twelve of our modelling experiments

We have also added additional text to *lines 271-273* (Methods section) to clarify our definition of the duration of the empirical shifts.

2. To support clarity, I'd suggest to add one or two models to your modelling section in the paper where regime shifts are specifically addressed. Currently, the WSP model is the only one that lends itself for testing with the breakpoint function. I'd suggest to add one or two models that address specifically ecological regime shifts, e.g. forest to savanna and eutrophication. Many of these models already exist, see for example Scheffer's book on Critical Transitions in Nature and Society (2009), or Solé book on

Phase Transitions (2011) — both have useful appendixes with various models, some of which are spatially explicit. I believe the WSP is the only model with a first order critical transition, while the GoL and LC seems to be second order critical transitions: they are continuous in the first derivative but discontinuous in the second. That’s why you are “forced” to use time spend in regime one as proxy of duration, but most importantly for their implications on real regime shifts, GoL and LC does not seem to have hysteresis (or correct me if I’m wrong). The motivation for two extra models is to allow yourself to test the shift duration as the time from crossing the threshold to realising the alternative regime, which is what the breakpoint function tries to get. Additionally, having a in-build threshold on a slow variable also allows you to test duration in the case of hysteresis, when the path to recovery differs from the path to collapse in the bifurcation. Is the duration the same when hysteresis is at play?

We have added two models to the manuscript that are consistent with the reviewer’s recommendation (bringing the total number of models up to five). The two additional models both explicitly include ecological regime shifts driven by positive feedbacks and tipping points. As suggested by the reviewer, one of these models, *Spatial Heterogeneity*, is taken directly from Scheffer’s book on Critical Transitions in Nature and Society (2009). Our results when hysteresis is at play in both of the new models are consistent with our previous models (Fig. 3 and Fig. S11). Note, however, that we did not use “time spend [sic] in regime one” as a proxy for regime shift duration – we use the time taken to shift from regime one to regime two. This assertion arose because the previous draft was not very clear on this definition (see previous comment), and we have consequently made the above changes (see response to previous comment) to hopefully clarify any confusion.

3. I would also like to encourage you to interpret your key finding. A power law is a statistical pattern, but it would be very useful for the reader (and the impact of your paper) if you can provide an informed speculation of why that pattern emerge? What do you think is underlying that 0.8 slope on the log-log scale? While I was reading your paper I was also reading the book *Scale* by Geoffrey West who finds many power law distributions in a number of different systems —they are very important features of complex systems— but also guides the reader on why they emerge, what do we know so far about their generative mechanisms or what are we missing. I believe a few sentences on the question why, even if speculative, can inform future hypothesis to explain that pattern.

We apologise that our interpretation of this pattern was not clear. We have reviewed our discussion section in light of this comment and include interpretation of our main findings in terms of real systems (*lines 120-134*) and in the section **Mechanisms in real and model systems** (*lines 165-205*). We had explicitly cited and discussed our results in relation to Geoffrey West’s work on *lines 199-205*.

Specific comments

Line Comment

All minor comments suggested by the reviewer have been addressed.

4. Lines 51-52: I did not understand “why in reverse”. Could you expand a bit?

By ‘in reverse’ we meant that systems should scale sub-linearly during their collapse. This has now been clarified in the text (*lines 62-67*).

5. Line 98: It is hard to conceptualise clear regimes in the GoL, perhaps some live cells vs all death. But then it is irreversible and does not have hysteresis. How these

models compare with the cases reported in your empirical analysis? They seem to be reversible, at least some of them on human time scales.

Our manuscript contains a broader range of regime shifts, defined by “large, persistent, and often unexpected changes in relatively stable ecosystems, which may (or may not) be driven by reinforcing feedback loops beyond ‘tipping points’” (*lines 32-34*). This is because the irreversibility and hysteresis of our empirical data are sometimes not known. Our models fit this same definition (i.e. all showing a shift from regime one to regime two without judgement of irreversibility nor hysteresis [although our two new models LCH and SH do both demonstrate hysteresis]).

6. Line 127: Doesn’t this contradict the network finding. GoL with 4 or 8 neighbours is the same as a homogeneous network with degree 4 or 8.

We apologise if the interpretation of our results was not clear here. We find that increasing the number of GoL neighbours leads to a quicker shift (Fig. 3, GoL-2.3), but there is an insignificant relationship between regime shift duration and the number of network connections (Fig. 3, LC-3.2). We discuss the reasons for these two results in *lines 152-158*, adding new text and references (*line 155*) to clarify the earlier confusion.

7. Lines 132-34: This is a bit of an overstatement — if I understood correctly the operationalisation of fluidity in the modelling exercise, it is defined as how far a disturbance spreads. You are assuming that the disturbance in water is more fluid than in land. But that is not necessarily the case. Climate change impact both at large scales, but other more localized perturbations might follow similar patterns of disturbance in land and sea i.e fishing and deforestation — both require mobility and humans are constrained by place and tech to move.

Essentially, we are offering a hypothesis to explain why two different simulation models may produce slightly different results. We accept the reviewer’s comments here and have worked to clarify the text by making it explicit that we are offering a hypothesis (*line 152*), as well as adding texts to *lines 156-158* to support our argument.

8. Lines 143: “...systems of all kinds transmits stresses more slowly through greater distances and absolute modularity” — again, it depends of the disturbance. Another aspect not discussed in the paper is the time scale of the species involved. In marine systems primary produces are often plankton (generation time of weeks), while terrestrial systems are slow growing species (trees generation times of centuries). Thus, it might not have to do with the duality terrestrial vs. marine, but rather the time scales of the species involved.

We highlight a first order approximation of the spatial dynamics underlying regime shifts and also indicate that system fluidity as well as system size may impact the speed of transitions. The reviewer’s hypothesis that generation time of species in the system may be important is likely true. However, unfortunately data deficiency prevents us from asking this question within our manuscript.

9. Line 151: Be careful with this statement: the existence of key nodes or influential nodes depends on network structure. For the purpose of disintegrating the network, targeting hubs is effective in scale-free networks. For the purpose of changing the state of notes, hubs are not necessarily the best and it also depends if the speeding process underlies a simple or complex contagion dynamic. For small-world networks, it depends where the network is in the spectrum of high-low clustering, not all small-world networks can be controlled with high degree nodes.

We agree with the reviewer that our previous assertion could be read as an overstatement. As such, we have rewritten the interpretation of our network results (*lines 185-195*) and bolstered the introduction and rationale around the network analysis in *lines 60-64*.

Supplementary information:

10. Line 38: Does this include time spend on regime 1 or time since tipping point crossed? Since the empirical data does not necessarily have information about thresholds and when they were crossed, how was the duration measured? Did you used the same breakpoint function on the time series of the papers selected?

This comment is similar to the first comment made by Reviewer 1. We apologise that the definition of 'regime shift duration' was not entirely clear in the initial submission. As such, we have added extra text to *lines 34-35* to clarify our overarching definition, added more detail in *lines 271-273* to clarify the methods used to measure the empirical regime shift durations, and given detail of both duration measurements and use of breakpoint functions in the section (*lines 589-516*) **Identifying regime shift durations of modelled time-series.**

11. Line 103: I didn't understood well enough the size of modules experiment. I'm assuming a module is the height of the landscape, and then the sum is necessary for comparison purposes, but also means that there is series of say 2-cells landscapes glue together but that do not influence each other. Is that right? If they do influence each other, how is the experiment different from the size of landscape — I guess what is not clear to me is what is a module on the GoL. I understand a module as a unit of a landscape or a network with more interactions than with its surroundings, but it is still connected to its surroundings.

We apologise that our definition of a module was not perfectly clear in the initial submission. As such, we have worked to fix this by adding a new graphical representation of modularity in *Fig. 1*, as well as additional explanatory text to *lines 54-57*. It is now clear that the modularity experiment modelled independent and unconnected sub-systems (*lines 54-57*), and that the regime shift durations equalled the sum of the individual sub-systems (*lines 376-390*).

12. Lines 149-51: Why is it the standard deviation a good measure of heterogeneity? If the underlying distribution of degree is normal (Gaussian) then it's a good approximation, but for other degree distributions such as small-world or scale-free, it can be misleading. In heterogeneous networks there is a negative correlation between in- and out-degree. A common measure is $\langle |k_{in} - k_{out}| \rangle / \langle k \rangle$ which lands in the range 0-1 and is easy to interpret.

As the reviewer highlights, there are many ways to mathematically measure the variance in the number of connections of the LC model (e.g. range, interquartile range, variance, standard deviation, standard error, 95% confidence intervals and so on), but when the underlying distribution is normal then standard deviation is a good approximation. We now confirm that the underlying distribution of our language change networks is normal (*lines 420-421*).

13. Lines 169-70: There is no Table 2 in the paper (results from a GLM?), and when omitting observations in a regression, its a good practice to state how many observations were omitted.

Table 2 in the previous submission was an erroneous label; this has now been fixed to Fig. 3 in the latest submission. We have also added the number of omitted model runs to SI 2.

However, it must be noted that omitted model runs should not be categorised as omitted observations. All cases where a regime shift was observed were included in the models, only those model runs without this observation were excluded (i.e. if the model run did not go through a regime shift then we are unable to use those data to investigate the speed at which the shift occurred).

Reviewer #2 (Remarks to the Author):

In the submitted manuscript the author(s) examined the empirical relationships between size and transition duration of systems that experienced regime shifts and used three different simple models to demonstrate the role of system fluidity, connectivity and modularity in causing such relationships. The paper is clearly written. I find this work very interesting and highly relevant to the hot topics critical transitions and ecosystem resilience. To my knowledge this relationship that was examined in this study has not been explicitly explored, presenting clear novelty. However, I have major doubts about the models, and feel that there are some concepts to be further clarified and additional analyses needed.

1. The author(s) focused on ‘regime shifts’ or ‘tipping points’ or ‘critical transitions’ in (socio-ecological systems. These system behaviors of abrupt changes have been mostly linked with alternative stable states (ASS) whose mathematical underpinning is fold bifurcation. Indeed, the empirical database used in this study contains a range of systems that have been suggested to have ASS (and fold bifurcation). For any real-world system it is difficult to give solid proof that ASS underlies system behavior, however, in the modelled system this is doable in an explicit way. My major concern is that none of the simple models have been shown to have ASS. And in my opinion most (if not all) models would not have ASS as the essential element of positive feedback is largely. I am well aware that one can argue that real-world regime shifts are not necessarily underpinned by ASS, and there are some other types of bifurcations other than fold bifurcation that may be the mechanism of the observed abrupt changes. Nonetheless, there is a clear mismatch between the real-world and modelled system behaviors in the sense of ASS vs. non-ASS. I think it is critical to include a couple of simple models that have been extensively used for the study of ASS such as the classical May’s grazing model and some more recent models used for unravelling the collapse of pollination networks or social networks. I understand that would mean a lot of additional analyses, but the current modelling results are not really convincing.

This comment is similar to the first comment of Reviewer 1. As such, we have conducted the additional analysis using two models (LCH and SH) of similar specification to those suggested by the reviewer. Critically, both of our additional models include spatial terms, thus allowing us to control for the effects of system size and fluidity on the duration of regime shift, and both have been demonstrated to have alternative stable states and hysteresis (see *Methods* and Fig. S11). The details of the new models are housed in the *Methods* section, whilst *Table 1* describes their experimental setups and *Fig. 3* displays the resulting associations between the spatiotemporal dynamics of regime shifts. Moreover, we have been careful to clarify our definition of regime shifts in *lines 32-35*.

2. For systems that have bifurcation behaviors, it would be great if the author(s) could systematically analyze the transient states if they simply focus on the duration of transition. But I feel it’s also valuable if they could also perform systematic analysis on how systems size affects resilience because the duration of transition is also

influenced by system resilience (depending how you define resilience) other than transient states when facing complex perturbation.

Whilst we agree that the exploration of resilience in all its many forms will make for an interesting analysis, we feel that this recommendation is beyond the scope of the current paper due to data deficiency. Put simply, we do not know how resilient our empirical systems were nor the strength of the stresses they were under. Similarly, three of the five models start unstable, and so we are not able to model the resilience of the system prior to the transition. Thus, our primary focus is to present the first attempt to analyse the relationship between spatial and temporal scales of shifts, and the interior system structures that may inhibit or accelerate the shifts.

3. I would like to see analysis on the empirical relationship between size and duration of transition for different types of systems separately. It looks that they would have pretty different slope?

We have added this analysis (Fig. S4) and described the results (including possible reasons for the different slopes and related caveats) in *lines 159-163*.

4. While the observed relationship between size and duration of transition is pretty clear, important caveats should be mentioned when it comes to inferring real-world situations by extrapolating the relationships, as they are plotted on a log scale, the predicted duration of transition could have a very large range (as also pointed out by the authors). It brings large uncertainties for interpreting and predicting real-world transitions.

We have worked in a number of ways to better communicate the uncertainties associated with our projections of real-world transitions:

1. We have updated *Fig. 2* (Fig. 1 in the previous submission) to include the empirical regression model with its 95% confidence interval – allowing the reader to visualise uncertainty in the empirical dataset across the range of system areas
2. We have retained the 95% confidence intervals for the projections of the Amazon and Caribbean coral reef regime shift durations (*lines 123 and 129*, respectively)
3. We have conducted a two-stage sensitivity analysis (*Methods*) that shows the empirical sub-linear trend line is very robust to uncertainty in our data (*Lines 103-111, Fig. S5 and Fig. S6*).

Reviewer #3 (Remarks to the Author):

This paper addresses an interesting question, which is: how does the speed of ecosystem collapse vary with system size? The paper uses information from 42 different ecosystems (assembled for existing data bases) to address that question. They show that the time to collapse increases with system size. So, larger systems collapse slower than smaller one, but the authors also show that they do it disproportionately faster than smaller systems. The authors then use 3 models to investigate the role of ‘system fluidity’, ‘connectivity’ and ‘spatial modularity’ in driving the speed of system’s collapse: a wolf-sheep predation agent-based model, a game of life cellular automaton, and a language change agent-based model.

1. The data presented is interesting, but the authors don’t really highlight the practical relevance of their main findings about the dependency of speed of collapse with size. The fact that system size influences ecosystem collapse in expected. How does the

nonlinearity of the relationship change our understanding and predictive ability of ecosystem resilience to perturbations?

We have extended and reorganised our 'Results and discussion' section to address the reviewer's concern. We now address the practical relevance of our main findings in two focused sections: (i) mechanisms in real-world systems, and (ii) implications for governance. In turn, we explicitly discuss "how does the nonlinearity of the relationship change our understanding and predictive ability of ecosystem resilience" in *lines 201-208* and *lines 225-232*.

2. Their results are also insufficiently supported by robust statistic (in particular whether the trend of that relationship linear or nonlinear)

Acknowledging that this comment is key to communicating the importance of our results, we have worked extensively to improve the statistical robustness of both the empirical and modelled relationships:

1. The Monte Carlo sensitivity analysis (*Methods, Fig. S5 and Fig. S6*) provides a further statistical assessment of the robustness of our empirical relationships between system area and shift duration.
2. The first section of our supplementary information (*SI 1*) analyses the key differences between the linear and non-linear relationships, finding that the sub-linear empirical relationship is the most robust and so is the only data we include in the new main text.
3. We conduct the equivalent linear versus power law analysis for our modelled data (*Table S8*), allowing readers to see the circumstances where a linear trend line performs better than a power law relationship.

3.I am also not convinced by the modeling part: the choice of these minimal models is not justified and seems odd since none of them is related to the type of data analyzed; the model analysis is also not thorough enough to really be convincing.

As detailed above, we have added extra modelling activities and analysis in the form of the Lake Chilika fishery system dynamics model (Cooper and Dearing 2019) and Scheffer's (2009) Spatial Heterogeneity model. As such, the five models in our study should now cover the range of regime shifts types found in reality, from gradual transitions from one state to another, to feedback-driven transitions with tipping points and hysteresis. The regime shifts observed in our models are comparable to those observed in our empirical data due to a shared definition of regime shift (now clarified in *lines 32-35*).

I here below list some major concerns and some more minor comments on the manuscript and the supplementary information.

Major concerns:

**4.The relevance of the main result is not sufficiently discussed. Larger systems shifts more slowly than smaller systems but disproportionately faster. The first part is very much expected. The second part may be more interesting, but:
- how robust is that trend? Based on the figures and on the numbers in the text, the differences seems minimal. Unless I missed it, the authors did not show a statistical test of the comparison between the different fits.**

Our results are extremely robust. We observe a sub-linear trend across all our sensitivity analyses (i.e. across all 42 alternative models and when data uncertainty was simulated [*Lines 103-111*]). We also refer to our response above to the second comment by Reviewer

3, regarding the addition of a dedicated section for the comparison of the linear versus sub-linear trends and their relative robustness (SI 1).

5. The authors don't mention anything related to the uncertainty behind the data points, but based on the proximity of the two fits, couldn't it be that this result becomes not significant when taking into account all the uncertainties behind the data? Even if the difference between the two fits is statistically significant, and the nonlinear fit is indeed better, how relevant is that difference (in magnitude)? The estimates of time to shift seem to be really close in both cases, and again could well be within the range of uncertainty.

We added a Monte Carlo analysis (see *Methods, Fig. S5 and Fig. S6*) to explore whether the sub-linear relationship remains across the range of uncertainties behind the data. This new sensitivity analysis shows our results are robust to uncertainty within our data. More information is given in our response to comment #4 by Reviewer #2.

6. What are concretely the implications of such results? Why should this result affect the way we look at these ecosystems? We live in an increasingly fragmented world; what do these results mean in that context?

With regards to the question "we live in an increasingly fragmented world; what do these results mean in that context?", we have added text to explicitly refer to these questions in lines *lines 167-171* and *lines 213-218* and more generally within section (lines 165-205) **Mechanisms in real and model systems** and section (*lines 207-221*) **Implications for governance**. Please also refer to our answer to comment #1, Reviewer #3.

7. The data is insufficiently presented and discussed. It seems to assemble very different studies, conducted by different people, at different scales, with different methodologies, for different purposes. What's the error on the estimation for times to shifts? Maybe this does not matter too much since the trends exists across orders of magnitude, but it would still be important to mention and discuss this.

Given that the empirical data is secondary (i.e. collected from previously published studies), it is very difficult (if at all possible) to estimate the errors underlying the empirical times to shift, owing to a lack of information in the original publications. Therefore, our Monte Carlo sensitivity analysis (*Methods, Fig. S5 and Fig. S6*) takes a more *post-hoc* approach, analysing the robustness of the sub-linear trend over a wide but plausible error range (i.e. 50-150% of the original shift duration values). Please also see our response to comment #2, Reviewer #3 for more details.

8. I am puzzled by the choice of the models. These models really don't seem the best choice to test the author's idea. Why choosing these models? In what way are they complementary? Why not using actual ecosystem models and maybe also socio-ecological models? Also the presentation of the models lacked information about the types of behaviors that the models exhibit: What are the dynamical behaviors of these models? How do they behave along gradients? What makes them shift? Some of this information is provided in supplementary information, which is well written and clear. But some of the basic information would need to be moved to the main text for the reader to be able to understand the results. For example, I did not understand what the authors meant by 'module' and how this was implemented concretely in the simulations.

We have made four changes to address these concerns:

1. As detailed in our response to comment #2, Reviewer #1, we have added two additional ecosystem models (LCH and SH) that include positive feedbacks, tipping points and hysteresis.
2. We have also added text to further justify the chosen models (*lines 73-77 and lines 83-90*).
3. In accordance with comment #14, Reviewer #3, we have clarified a number of terms in *Table 1* (including adding the column 'model type') to better describe the dynamical properties and parameterisation of the models used.
4. As detailed in comment #11, Reviewer #1, we have clarified our definition of modularity by: (i) adding a graphical representation of modularity in *Fig. 1*, (ii) adding explanatory text to *lines 54-57*.

9. This is only a suggestion, but the results that I found the most interesting were the actual quantitative values given for times to shift in various ecosystems (l. 13-15). Maybe a more convincing way of presenting the results of this analysis could be to first mention that result (i.e. the fact that across ecosystem types, it seems that ecosystems are likely to collapse at years to decades timescales, meaning much faster than expected?), then mention the dependency of time to collapse to system size, and finally discuss the problem of habitat destruction/fragmentation in that context?

We have adjusted the structure of the manuscript based on this comment – the real-world implications are now at the forefront of our results/discussion (*lines 120-134*). For example, we now describe how our sub-linear model means that regime shifts occurring at the spatial scale of some of Earth's most iconic ecosystems should "remain within 'human' timescales of years and decades" (*line 133*). We then discuss our modelled results and some of the potentially contributing spatial structures, before discussing the implications of our results for complex systems (*lines 165-182*) and contemporary social systems (*lines 198-205*).

Minor comments

10. lines 5-6: Is the frequency of regime shifts predicted to increase? Are there any formal study really demonstrating this or is this just a hypothesis based on a verbal argument?

Here we have added a supporting reference (*reference #14, line 43*) in the form of Drijfhout et al. (2015, *PNAS*), who projected an increasing number of climatic and biophysical 'abrupt shifts' with the Earth System Models used by the Intergovernmental Panel on Climate Change (IPCC).

11. lines 58:65: Move this part later in the text? This is a bit too detailed at this moment in the text. It would go better around l. 80.

We agree with the reviewer that the explanations of our hypotheses came too early in the previous submission. Therefore, we have shortened, simplified and merged the justifications for the two hypotheses (*lines 24-67*) and added *Fig. S1* to help visualise the complex systems concepts that inform our predictions.

12. lines 70-71: 'the overarching relationship is sub-linear': I guess that you tested that this fit was significantly better than a linear relationship? I might have missed it but this information does not seem to be clearly mentioned in the text and the statistics for the comparison of the fits are not shown (or I did not find them). Even if that's the case and a sublinear fit is indeed better than a linear fit, it's not really clear that this makes a significant difference for the estimation of transition time (see *Fig 1* – I realize that the scale is log-log). It seems on the contrary, that it does not matter

much except for very small systems? This should more thoroughly explained and discussed.

We have now added a section to the supplementary information (SI 1) which analyses the differences between the linear and nonlinear fits (both in terms of their predictions and statistical robustness), as well as table S8 which compares linear and nonlinear fits for the modelled data. Our response to comment #2, Reviewer #3 also addresses this issue.

13. lines 83: It is not clear at this point of the text what those terms mean. Moving the text from l. 58-65 here might help. We also need more detailed information: what parameter of the model were varied concretely?

Unfortunately, we are not clear which 'terms' the reviewer was referring to in line 83 of the previous submission. Regarding the second part of the comment, we have redesigned *Table 1* to be more explicit about the model parameters varied (and their parameter value ranges).

14. Table 1 is very difficult to follow. The headers of the first row should be changed to more explicitly describe their content.

Col 1: should mention the type of model (mathematical formalism: ODE, PDE, CA....)

Col 2: 'Experiment' → 'parameter varied'?

Col 3: very redundant with col 2: merge?

Col. 5: 'repeat' → replicas? Repetitions?

Col 4: 'runs' → 'simulation length'? 'number of timesteps'? Or is this the number of runs? In any case, there is not enough information in this table for us to understand what this means. Actually I would consider moving this whole table to appendix.

We have made all of the recommended changes to Table 1 and the table caption now refers the reader explicitly to the Methods section and SI boxes for more information. However, we have not removed the table from the main manuscript, as we feel it is important for the readers to refer back to it when visualising the modelling outputs in Fig. 3.

15. lines 90: What does 'dynamic spatially distributed system' mean concretely? What kind of mathematical formalism does that correspond to?

We agree with the reviewer here that the previous text was not clear; as a consequence, we have removed this text from the manuscript, and instead inserted the column named 'model type' to communicate the modelled system more clearly.

16. lines 96-97: Are these alternative stable states?

The answer to the reviewer's question here is 'yes'; however, we acknowledge that the previous text was not clear in this point. Please see our response to comment #1, Reviewer #1 regarding the steps we have taken to clarify our definitions of regime shifts and their duration.

17. Figure 2 is difficult to understand. It would really help to have the names of columns (use the same name as in the text and table 1 for the 3 parameters varied) and rows (i.e. names of models). Along the x-axis: it would help to use the same terminology as in the text. For at least 6 of the 9 panels, it's very unclear whether linear or non-linear fit are better. It would be nice to show the outcomes of a statistical test.

What is now Fig. 3 has been made clearer with above recommendations. Fig. 3 is also now linked to Table 1 with the different experiment names. Also, Fig. 3 has become visually clearer, as we now only plot the log-log linear relationships, with the linear relationships from

the unlogged data plotted in Fig. S1. Furthermore, we compare the empirical fits in SI 1 (i.e. linear versus non-linear), and the modelled fits in Table S8.

18. lines 144-145: Is that seen in the data?

The reviewer is referring to the following statement from the previous submission: “*The finding that larger systems take disproportionately less time to shift between regimes, relative to their size, is consistent with the empirical findings*”.

We have revised our descriptions of the modelled results to better flag the consistencies (and discrepancies) with the empirical data (*lines 138-142*).

19. lines 156: What is 'self-organisational energetics'?

We have changed the text to say ‘self-organisation’, which is a commonly accepted term within the broad fields of ecology, sustainability and complex social-ecological systems (now *line 168*).

20. lines 173 and following: These estimates are interesting, but how do you take modularity and connectivity of the ecosystems into accounts? Do you suppose these whole ecosystems are one homogeneous fully connected system?

Our first-order projections for the Amazon and Caribbean coral reefs are based on their area (and of course the areas of the other systems, which make up the empirical model detailed in *Fig. 2*). As such, we do not make any assumptions about the homogeneity or connectivity of the system. However, we discuss the implications of these spatial characteristics in our **Mechanisms in real and model systems** section (*lines 165-205*).

21. lines 186-187: This sentence seems to contradict the network literature on modularity: modules are usually thought to confine the spread of perturbations in complex systems, and they are therefore usually found to be stabilizing (which I would expect should lead to longer shift duration).

The reviewer’s comment is correct and agrees with the findings of our paper (specifically the two modularity experiments, WSP-1.2 and GoL-1.2). This confusion arose as these lines were not clear. To clarify, we find that increasing modularity *does* indeed lead to slower shifts, however, there is a law of diminishing returns as modularity continues to increase (i.e. the sublinear trend). See *lines 192-193* and *lines 212-218* for our clarification.

22. As mentioned, I had a hard time understanding what was meant by ‘modules’ in this study, isn’t it rather habitat fragmentation? (it’s not really that the structure is modular but more that it is patchy). In the SI, for the Wolf sheep predation model, I did not understand what was meant for the modules. The SI mentions (l. 70 of SI) changing the height of the landscape but does not that correspond to changing the size of the landscape?

In accordance with comments from other reviewers, we have taken numerous steps to clarify our definition of modularity (both graphically and in text). For example, see our response to comment #11, Reviewer #1.

23. line 206: ‘heightened risk of current activities triggering sub-global tipping points’: what does that mean? And what is the evidence for that? This paper does not seem to provide elements to support such a claim.

The line quoted in the reviewer's comment was not actually a finding from our study, but an implication from the three studies cited in the previous clause, which amplify our main finding (i.e. that once triggered, large systems may shift over timescales that are relatively short for their size). However, we acknowledge that the wording of the previous text was confusing. The re-phrased text in *lines 222-226* should clarify any confusion.

24. line 207: 'with the findings presented here, humanity now needs to prepare for even faster change in shifting ecosystems': again, I don't see how this is in agreement with the results presented here. In what sense do the results suggest an acceleration of shifts?

We again apologise that our wording was confusing. Our results do not suggest that shifts for individual systems will accelerate (as in, systems will collapse towards their alternative state at an increasingly faster rate). Instead, our results imply that in comparison to linear scaling between system size and shift duration, regime shifts in larger systems may take less time than previously envisaged. The new text in *lines 226-229* should clarify this conclusion.

25. Again, in the SI, regarding the Game of life model (l. 88 of the SI), what does it mean that the model is 'inherently unstable' when cover is below 35%? That the system goes to global extinction? L. 90-91 seems to contradict that. That there are cyclic dynamics? Again for this model, I did not understand the meaning of modules.

We have added text to the Methods (*line 361-368*) and two extra figures to the supplementary information to clarify what is meant by model instability (Fig. S9 and Fig. S13). Please see comment #11, Reviewer #1 for our response to reduce the confusion around the definition of modules.

Reviewers' Comments:

Reviewer #1:

Remarks to the Author:

The authors have done a good job in clarifying what has been done and why it is relevant for the broader scientific audience. The manuscript is well balanced, well written, and enjoyable to read. The definition of shift duration is now more clear and consistent throughout the paper. My only concern at the moment is the quality of the figures which do not do justice to the paper. Since the main result of your empirical and theoretical analysis is the exponent of the power laws, or the slope of the curves fitted in the log-log space, both the x and y axes should be equivalent. That is, the 1:1 line should have 45 degree (slope = 1) and intercept at the origin (0,0). Which is not the case, especially in Figure 3 where you compare multiple models but x and y axis vary from panel to panel (and sometimes the reference line had negative slope?).

Below a few specific comments that I hope improves the clarity of the paper.

Specific comments

Lines Comment

151-54 This long sentence reads a bit strange ... "stressor has greater freedom to choose its direction..." — In nature stressors just happen (fire, disease transmission), they don't have choices.

177 The hidden feedbacks reported in reference 37 are not all positive. I suggest delete the word "positive"

179 The claim "the greater modularity in large systems effectively accelerates this process" contradicts your result reported in line 146 "modular systems are slower to shift from one regime to another". On the example that follows in line 180 about the fire, it is the opposite: more frequent fires reduces the probability of fires because it consumes the available fuel. More frequent fires do not leave enough time for dry matter to accumulate.

211-12 Could you please elaborate, this sentence sounds counterintuitive, if not contradictory. If I understood correctly, you're saying that something that confers resilience against collapse could at the same time amplify the collapse?

Reviewer #2:

Remarks to the Author:

I think the authors have done a nice job of improving the ms. I like the additional model analyses and revision. I have no further comment and recommend for publication in Nature Communications.

Reviewer 1 – major comments

1. The authors have done a good job in clarifying what has been done and why it is relevant for the broader scientific audience. The manuscript is well balanced, well written, and enjoyable to read. The definition of shift duration is now more clear and consistent throughout the paper. My only concern at the moment is the quality of the figures which do not do justice to the paper. Since the main result of your empirical and theoretical analysis is the exponent of the power laws, or the slope of the curves fitted in the log-log space, both the x and y axes should be equivalent. That is, the 1:1 line should have 45 degree (slope = 1) and intercept at the origin (0,0). Which is not the case, especially in Figure 3 where you compare multiple models but x and y axis vary from panel to panel (and sometimes the reference line had negative slope?).

We have made the suggested changes for each of the 12 panels in Figure 3 and adapted the text in the caption of Figure 3 to reflect the changes. It should be noted that three of the panels have a negative 1:1 line (origin of 0,0 and gradient of -1), which is consistent with the modelled results [see lines 144-146].

Review 1 – line comments

1. 151-54 This long sentence reads a bit strange ... “stressor has greater freedom to choose its direction...” — In nature stressors just happen (fire, disease transmission), they don’t have choices.

Please see changes to text from lines 160-161.

2. 177 The hidden feedbacks reported in reference 37 are not all positive. I suggest delete the word “positive”

Edit made as suggested [line 187].

3. 179 The claim “the greater modularity in large systems effectively accelerates this process” contradicts your result reported in line 146 “modular systems are slower to shift from one regime to another”. On the example that follows in line 180 about the fire, it is the opposite: more frequent fires reduces the probability of fires because it consumes the available fuel. More frequent fires do not leave enough time for dry matter to accumulate.

The confusion arose as line 179 was not clear. The second quote in the reviewer’s comment refers to the overarching relationship between modularity and shift duration (i.e. increasing modularity leads to longer shifts, in general); however, the first quote refers to the sublinear relationship (i.e. the ability of increasing modularity to further slow down shifts weakens as modularity increases). We have changed the text from line 189 to clarify our point.

4. 211-12 Could you please elaborate, this sentence sounds counterintuitive, if not contradictory. If I understood correctly, you’re saying that something that confers resilience against collapse could at the same time amplify the collapse?

We have added additional text from line 222 to illustrate our point.